# Perfect Impedance Matching with Meta-Surfaces Made of Ultra-Thin Metal Films: A Phenomenological Approach to the Ideal THz Sensors

**DOI:** 10.3390/ma13235417

**Published:** 2020-11-28

**Authors:** Binglei Zhang, Yang Liu, Yi Luo, Feodor V. Kusmartsev, Anna Kusmartseva

**Affiliations:** 1Micro/Nano Fabrication Laboratory, Microsystem and THz Research Center, Chengdu 610200, China; zhangbinglei@mtrc.ac.cn (B.Z.); liuyang@mtrc.ac.cn (Y.L.); 2Physics Department, Loughborough University, Loughborough LE11 3TU, UK; A.Kusmartseva@lboro.ac.uk; 3College of Art and Science, Khalifa University, Abu Dhabi P.O. Box 127788, UAE

**Keywords:** terahertz waves, THz devices, optical, impedance matching, meta-surfaces, anti-reflection, metal nanofilms, oxide layers

## Abstract

The terahertz (THz) frequency range is incredibly important as it covers electromagnetic emissions typical for biological and molecular processes. All molecules emit THz waves in a unique fingerprint pattern, although the intensity of such signals is usually too weak to be detected. To address the efficiency gap in existing THz devices it is extremely important to create surfaces with perfect anti-reflection properties. Although metals are absolutely reflective, here we show both theoretically and experimentally that by constructing meta-surfaces made of a superposition of ultra-thin metallic nano-films (a couple of nanometres thick) and oxide layers a unique property of perfect transmission and impedance matching may be realised. The perfect transmission rates can be as high as 100% and it may be achieved in both optical and THz regimes. The predicted effect has been observed for numerous meta-surfaces of different compositions. The effect found here is expected to impact the renewable energies sectors, optoelectronic and telecommunication industries, accelerating the arrival of the sensors for the new 6G-technology. The phenomenon is highly relevant to all scientific fields where minimising electromagnetic losses through reflection is important.

## 1. Introduction

Recent research into antireflective coatings made of multilayer structures of dielectric films has shown that it is possible to obtain very low levels of THz light reflection [1], opening up a new direction in nano-photonics with a broad potential for diverse applications. Despite the substantial success in current approaches to decrease the THz light reflection, obtaining surfaces with 100% anti-reflection and transmission rates still remains one of the main obstacles in the construction of perfect optical devices such as flat Veselago–Pendry lenses. Current advances in 2D materials have demonstrated that fabrication of hybrid structures made up of graphene and superconductors can lead to a significant enhancement in light sensitivity [2]. Similar trends have been observed in graphene covered by a thin film of colloidal quantum dots, where extremely strong photoelectric effect provides enormous gain for photo-detection (about 10^8^ electrons per photon) [3]. Graphene grown on SiC and its composites has also been shown to have a strong photo-response and may be used to enhance the efficiency in solar cells [4,5]. On the other hand, to improve the detection of THz waves, which are normally quite low in intensity, creating surfaces with extremely low reflection and complete transparency to THz frequencies is absolutely crucial. By realising materials with perfect THz anti-reflection properties, the existing THz technology can be advanced to the next level where it would become possible to study the chemical and biological processes of individual molecules or to develop novel encryption-decryption protocols based on single photon detection in telecommunications.

Recent innovations in thin film technology have shown that heterostructures made of graphene and superconductors can be used effectively to amplify and generate THz waves [2,6,7]. Such heterostructures are unique objects that utilise the principles of vacuum solid-state nano-electronics for the amplification and detection of THz waves. In high-quality graphene monolayer, the electron mean-free path can be as long as several tens of micrometres. The ultrahigh electron mobility allows for ballistic electron transport and, therefore, leads to the creation of coherent electron circuits. The presence of such circuits may be exploited to establish coherent states triggered either by the proximity to a superconductor or the formation of a graphene p-n junction in a magnetic field [7]. In the case of a graphene p-n junction in a magnetic field the electrons are confined to move along snake trajectories. These are topologically protected states, where back-scattering is absent and, therefore, thermal losses are reduced to zero. Thus, the presence of such coherent states may facilitate the interaction of electromagnetic waves with electrons, and infinitely enhance the photo-response efficiency. This would result in a direct, loss-less transformation of energy from electromagnetic waves to electrical current and vice versa. Combining this effect with perfect 100% anti-reflective surfaces would enable the development of extremely efficient and sensitive THz detectors to be used in fundamental research as well as numerous applications, from studying remote galaxies, providing encryption in wireless communication, monitoring health and environment pollution to medical imaging and virus detection [8]. In order to achieve 100% anti-reflection and perfect transmission of THz waves a precise impedance and phase matching of light across a surface needs to be obtained.

It was Lord Rayleigh’s, who first discovered that tarnishing in aged glass (circa 1886) [9] created an interlayer with a refractive index much lower than glass leading to a much lower overall reflection. In general, modern technologies employ many anti-reflection methods (see, the recent review [10], for details). However, all existing real coatings so far have substantial limitations in terms of performance, where even in best cases the reflection coefficient is reduced to only about 0.1% or the transmission coefficient is 99.9%. Recent pioneering work on dielectrics has demonstrated that a thin dielectric film deposited on a metallic substrate can selectively absorb various frequencies of the incident light [11]. Because of the small thickness (few tens of nanometres) such dielectric nano-films have a low sensitivity to the angle of incidence and require only little absorbing material. The other significant disadvantage in current technologies is that frequently the anti-reflection properties are constrained to one distinct wavelength of light [10,12]. In order to remove the Fresnel reflection over a broad wavelength range, it is generally necessary to have an optical material where the refractive index gradually spans (or matches) across the connecting interfaces (e.g., from air to substrate) [9,10,11,12,13]. There are many techniques for producing systems with a graded refractive index matching to ambient air, such as chemical vapour deposition, rough surface and interference-patterning. However, despite success, in all cases reaching 100% transmission and anti-reflection over a broad wavelength range was still not possible, see the review [1] for a detail.

The impedance matching effect is very important for terahertz technology because naturally occurring THz waves are usually very weak. Therefore, any mismatch in impedance can stop their propagation through a medium, making them completely invisible. There have been many attempts to improve on the impedance mismatch in the terahertz range. The conventional approach to establish the impedance matching to vacuum, which is considered as a medium with the impedance Z_0_ = E_x_/H_y_ = 377 Ohms, is achieved by stacking dielectric layers to produce anti-reflection coatings. This is analogous to having Fabry–Perot resonators, where the impedance matching condition is satisfied by varying the thickness of a dielectric layer embedded between partially reflecting mirrors.

The research in the area has received a considerable boost since the pioneering discovery that hole perforation in metallic films can enhance light propagation. It has been shown that if the size of the nano-holes made in a metallic film is less than the wavelength of light, perfect transmission can occur [14]. Effectively, that means that such a perforated metal film has an impedance which is matching to a vacuum. This discovery has attracted a lot of attention, and the effect has been confirmed both for the terahertz and microwave ranges [15,16]. The intensive theoretical research that followed has predicted the existence of similar phenomena in a broad range of systems, which constitute lossless plasmonic materials. The behaviour is expected to occur in metallic gratings with very narrow slits [17,18,19], where the metallic channels are much smaller than the wavelength. Furthermore, the effect is predicted for one-dimensional periodically nanostructures metals [20], and metallic metamaterials with a high index of refraction [21,22]. Comparable properties are expected for optically thick metallic films with circular subwavelength hole arrays [23,24,25].

The established terahertz time-domain spectroscopy (THz-TDS) is an important state-of-the-art technique in physics, chemistry, biology and medicine [26]. The unique, otherwise hidden, features and characteristics of many materials become observable in the terahertz frequency range. Therefore, enhancing the efficiency and sensitivity of the THz-TDS methodology is highly desirable for a wide range of applications. The THz-TDS devices require broadband optical components to manipulate the path of the THz waves. The optical elements used in these structures (such as beam splitters, lenses and photonic crystals) should have proper impedance matching to enhance the efficiency and performance. The standard anti-reflection method based on using quarter wavelength dielectric layer is not suitable here because of the narrow THz bandwidth [27]. Instead, a wide range of plasmonic metamaterials specifically adopted to the THz ranges are utilised.

The predicted phenomena of THz sensitivity in plasmonic metamaterials have first been confirmed at specific THz wavelengths for arrays of square and circular holes punctured in metal [28]. Similar behaviour has been achieved for films with cuboid metallic inclusions [29]. Here, the invisibility of these materials to THz waves is linked to impedance matching to vacuum resulting in a nearly perfect transmission. The use of thin metallic films for the impedance matching of infra-red waves has already been confirmed [30]. Here, it was shown that a metallic mirror with arbitrary reflectivity can be fabricated. The effect may be related to the finite conductivity of such films. It has been shown that for superconducting NbN [31] and for VO_2_ [32] films deposited on a sapphire substrate, Fabry–Perot resonances completely vanish. The pioneering paper by Unterreiner et al. [33] demonstrates that ultra-thin metallic films can be used to achieve an efficient impedance matching at THz frequencies. The work shows that the reflection of THz pulses can be tuned by covering a dielectric interface with ultra-thin layers of chromium and tin oxide of varying thicknesses. This was the first observation of THz antireflection.

Complementary approaches in achieving broadband impedance matching, include nanomaterials. Remarkable photo-sensitivity has been reported in graphene-oxide films processed as a solution [34]. Interestingly, randomly stacked multi-layer graphene interfaces demonstrate similar behaviours [35]. It was shown that the amplitude of THz pulses reflected from the quartz/graphene/air interface changes with the incident angle and the layer number of graphene. Another work reports the efficiency of THz photonic crystals (PC) composed of layers of high-resistance Si and polyethylene terephthalate (PET) films [36]. A high transmittance and suppression of Fabry–Perot resonances for a wide range of incident angles (0–60°) and a relatively broad spectrum (0.26–0.55 THz) in both polarisations has also been observed in Si/PET. Thus, one may design a PC optimised for any frequency by controlling the spatial dispersions of the effective permittivity or permeability of the films. Such metacrystals are exhibiting a unique wide-angle impedance matching, in general. Numerous other types of PC have been investigated [37,38,39,40,41,42]. However, in all described cases it was not possible to achieve a transmission rate above 99.9%, indicating that optical losses could be minimised further for any given frequency range.

In the present paper we describe the perfect impedance matching, which is achieved by coating a few atomic layers of metal films on an insulating oxide layer. We demonstrate a phenomenological approach in achieving 100% anti-reflection and transmission at a specific frequency band by adjusting the combination and the thicknesses of the metal and the oxide. We show that the method is completely general and the effect can exist at all frequencies of light. We demonstrate that by altering the oxide + metal combinations and thicknesses any frequency range across the THz to visible light spectrum can be targeted. Furthermore, we establish a theoretical framework to predict such behaviour for all frequencies. Our findings are corroborated experimentally. We demonstrate that the role of the ultra-thin metallic film layer deposited on the surface of an insulating oxide is to create effective refractive index matching between the air and the system. This is analogous to impedance matching for electrical microwave signals [10,11,13]. The technology based on this new effect has significant potential for a variety of applications ranging from solar cells to optical sensors, progressively impacting the energy, space and communications sectors. For example, the sensitivity of optical devices with 100% anti-reflection can be high enough that they can detect individual entangled photons providing a new pathway to explore and drive quantum communications and quantum technologies.

## 2. Meta-Materials and Meta-Surfaces

Our meta-materials or meta-surfaces and devices are composed of three layers: a substrate, an oxide layer and a metal nano-film. The first layer is the thick substrate made of n-doped Si. The second layer is the oxide which is either 540-nm and 430-nm thick silicon oxide SiO_2_ or 60-nm thick sapphire, Al_2_O_3_. Finally, the third layer is the metal nano-film ranging in thickness between 1 nm and 10 nm. The nano-films have been made by using e-beam deposition directly on top of the oxide layer. The film surface morphology is very different, depending on the film thickness (as indicated in Figure 1 top view in yellow). The diagram of the experimental set up is presented in Figure 1. During the experiments, the triple layer samples were irradiated by several light sources—including a halogen lamp and various lasers to cover the visible light range. Light reflection and transmission were measured with the ellipsometry method. Notably, these triple layer systems demonstrate 100% anti-reflection and transmission over broad wavelengths. The effect arises because the ultra-thin metal nano-film combined with the oxide result is making the effective refractive index of the entire system exactly the same as that of air.

The light ray diagram through this structure is illustrated schematically in Figure 2a. To calculate the effective refractive index and the light reflectivity and transmission for such a triple layer setup we have used a transfer matrix method. The quantum mechanical approach is necessary to provide a proper description of the light-matter interactions and the optical conductivity σ(ω) of the metallic nano-films, which consist of one or a few monolayers of atoms. In one model such ultra-thin films may be considered as an array of interacting quantum dots. Once the optical conductivity σ(ω) of the nano-film is known, we can make a correct estimation of both the refractive index and the extinction coefficient for the triple layer system as a whole. Thus, we show that the 100% anti-reflection and transmission is the result of numerous reflection and refraction processes between the three existing interfaces (1) the metal nano-film, (2) the oxide and (3) the substrate.

## 3. Methods

Here we show that by tuning the thickness of the metal nano-film, we can significantly change the effective refractive index of the triple layer system and even get it to match the refractive index of air. In this process 100% anti-reflection may be achieved for the entire system. To describe the effect, we have used Fresnel theory, even when the film is ultra-thin about 1 nm thick. The reflectivity from the entire multilayer system depends on the layer structure, the thicknesses of individual layers and their optical constants. The surface of the triple layer structure shown in the Figure 2a can be viewed as an interface between air and an effective medium having effective optical constants, as shown in Figure 2b. In this case, the amplitudes of the electric and magnetic fields on the surface and the substrate are *E*_sur_, *H*_sur_ and *E*_Sub_, *H*_Sub_, respectively (see, Figure 2b). For this simplified configuration the reflectance can be calculated using the standard Fresnel equation. The amplitudes of the electric and magnetic fields on the surface, *E*_sur_ and *H*_sur_, are determined through transfer matrix theory. In this case, the amplitude of normal incident electromagnetic wave becomes:(1)$$\left(\begin{array}{c} E_{sur}\\ H_{sur} \end{array}\right)=\prod_{m=1}^{S}\left(\begin{array}{cc} \cos\left(\delta_{m}\right) & i\;\sin(\delta_{m})/\gamma N_{m}\\ i\gamma N_{m}\sin(\delta_{m}) & \cos\left(\delta_{m}\right) \end{array}\right)\left(\begin{array}{c} E_{sub}\\ H_{sub} \end{array}\right)=M\left(\begin{array}{c} E_{sub}\\ H_{sub} \end{array}\right)$$
where *m* is the index of the layers, going from 1 to S for a considered structure. Normalisation parameter, 1/γ is the intrinsic impedance of free space $1/\gamma =\sqrt{\left(\mu_{0}/\varepsilon_{0}\right)}$. Here *N*_m_ and δ_m_ are the refractive index and the phase shift of the *m*-th layer: *N*_m_ = *n*_m_ − i*k*_m_ = *H*_m_/(γ*E*_m_) and δ_m_ = 2π*N*_m_*d*_m_/λ. *M* is the matrix product of the 2 × 2 transfer matrices of the individual layers. The effective optical constant of the multilayer structure is defined as *N*_eff_ = γ*H*_sur_/*E*_sur_, where $\gamma =\sqrt{\left(\varepsilon_{0}/\mu_{0}\right)}$ is one over the intrinsic impedance of free space.

Using the definition of the matrix *M* given in Equation (1), the expression for the effective optical constant of the trilayer system can be written as
(2)$$N_{eff}=\left(M_{21}+M_{22}N_{Sub}\right)/\left(M_{11}+M_{12}N_{Sub}\right)$$
where the subscript of *M* indicates the corresponding component of the matrix. The newly introduced formalism can now be applied to the triple layer structure, composed of a Si substrate, a deposited oxide (SiO_2_ or Al_2_O_3_) layer and a thin nano-metal film, as shown in Figure 2a. In this particular case, the index *m* is assigned to 1, 2 and ‘Sub’ corresponding to the metallic film, the oxide (SiO_2_) and the Si substrate, respectively. Therefore, *N*_1_ and δ_1_ are the refractive index and the phase shift of the metallic film; *N*_2_ and δ_2_ are the refractive index and the phase shift of the oxide; *N*_Sub_ and δ_Sub_ are the refractive index and the phase shift of the substrate. The effective optical constant for such triple layer setup then becomes:(3)$$N_{eff}=\frac{i\;N_{1}\;\tan\left(\delta_{1}\right)+iN_{2}\;\tan\left(\delta_{2}\right)+N_{Sub}\;\left(1-\frac{N_{1}}{N_{2}}\tan\left(\delta_{1}\right)\tan\left(\delta_{2}\right)\right)}{1-N_{2}/N_{1}\;\tan\;\left(\delta_{1}\right)\;\tan\;\left(\delta_{2}\right)+iN_{Sub}\;\left(\tan\;\left(\delta_{1}\right)/N_{1}+\tan\;\left(\delta_{2}\right)/N_{2}\right)}$$
where *N*_1_, *N*_2_ and *N_Sub_* are the optical constants of the metallic nano-film, the oxide layer and the Si substrate, respectively. Here, for simplicity, we will consider only normal incidence of the electromagnetic waves.

The dependence of the metal nano-film thickness *d*_1_ = *λδ*_1_/2*πN*_1_ on the effective refractive index *N_eff_* found from Equation (2) then takes the form:(4)$$\frac{2\;\pi \;d_{1}}{\;\lambda }=\frac{iN_{2}\left(N_{Sub}-N_{eff}\right)+\left(-N_{2}^{2}+N_{eff}N_{Sub}\right)\;\tan\;\left(\delta_{2}\right)\;}{N_{2}\;\left(N_{1}^{2}-N_{eff}N_{Sub}\right)+i\left(N_{1}^{2}N_{Sub}-N_{eff}N_{2}^{2}\right)\;\tan\;\left(\delta_{2}\right)}$$

Analysis of the obtained equations indicates that in order that the thickness of the metal film *d*_1_ or *δ*_1_ remains a real value the refractive index of the metal film *N*_1_, satisfies the following constraint:(5)$${\left(n_{eff}-n_{c}\right)}^{2}+{\left(k_{eff}-k_{c}\right)}^{2}=R_{0}^{2}$$
where *n_c_*, *k_c_*, *R*_0_ are parameters with complex dependencies on all refractive indices *N_m_* and all phase shifts, *δ_m_* of the composing layers. Similarly, *n_ef_*, *k_eff_* are the real and imaginary parts of the complex effective refractive index, *N_eff_*. These parameters determine the manifold of real and imaginary values for the effective refractive index, *N_eff_*, where the condition for the perfect refractive index matching *N_eff_ =* 1 + 0 *i* can arise. Specifically, when the insulating multilayer structure is coated by a very thin metallic film, the possible values of *N_eff_* constitute a circle in the complex space centred at (*n_c_, ik_c_*) with a radius of *R*_0_. The only variable in these parameters is *N*_1_, which can be estimated using some realistic values for the dc optical conductivity, *σ*_0_, the scattering time, *τ* and the dielectric constant associated with localised charges ε*_L_*(*ω*) of the metal nano-film. For very thin films N_1_ may also vary with respect to its thickness.

For example, the helical dependence of the metal nano-film thickness, *d*_1_, calculated using Equation (4), on the triple layer optical constant, *N*_eff_ = *n*_eff_ + i*k*_eff_ in the (*d*_1_, *n*_eff_, *k*_eff_) space is shown in the Figure 3a. By changing the nano-film thickness the optical constant *N*_eff_ follows the helical trajectory shown in red, see Figure 3. This calculation illustrates that the optical constant of the film is largely determined by the nano-film thickness. Note that the projection of *d*_1_ on the (*n_eff_, k_eff_*) plane is a circle as shown in the Equation (5). The trajectory presented in the Figure 3 has been calculated for ultra-thin Pt film, with *N*_1_ = 3.20 − 4.3 *i*, deposited on 430 nm of SiO_2_ on Si. The incident wavelength was chosen as λ=500 nm.

Using the obtained manifold, which is just a circle, *S*_1_ (see Equation (5)), we can immediately find the parameters necessary for the metallic film to have 100% anti-reflection, which is given by the impedance matching condition *N_eff_* = 1 + 0 *i*. The trajectory of *N_eff_* (*N*_1_) parameterised by *N*_1_ is shown in Figure 4a. The parameters used in these illustrative calculations are given in Table 1. The trajectories of *N_eff_* (*N*_1_) for a Pt film, and two artificial metals M1 and M2 are shown in Figure 4a as red (dashed), blue (dashed) and orange (solid) circles respectively. The precise values of *N*_1_ can shift the position and change the radius of the circular trajectory. Importantly, the crossing point between the trajectory circle and the horizontal axis at (1,0) shown in the Figure 4a, corresponds to the condition for perfect 100% anti-reflection. For example, with a hypothetical artificial metal M2, one possible solution occurs at *N*_1_ = 0.86 − 4.3 *i*, which is shown by the orange circle in Figure 4a. For this specific case, the thickness of the M2 nano-film, as calculated by Equation (4), is found to be about 3.12 nm. Calculating the reflectivity using the transfer matrix method gives a value of 0.025%, corresponding to a transmission of 99.975%. The calculated dependence of the reflectivity on the thickness of the M2 nanofilm, *d*_1_ is plotted in Figure 4b. The results show that to achieve an ideal anti-reflective coating, we must use nanometre-thick metal films. These calculations also demonstrate that the desired index matching frequently occurs for films with a small refractive index but a large extinction coefficient. Such characteristics are usually found in metallic nano-films. The present work shows numerous examples of 100% anti-reflection obtained in triple layer structures with nano-films made from different metals, experimentally corroborating our phenomenological model and its predictions.

## 4. Results

In order to evaluate the predictions of refractive index matching or 100% anti-reflection and transmission in metal nano-film capped multilayer structures, we have designed, fabricated and tested a series of devices or samples. The ultimate goal was to obtain a THz wave detector with a considered structure SC/GR/I/M, where (M) stands for the metal ultra-thin nano-film, (I) is an insulating oxide such as SiO_2_ or Al_2_O_3_, (GR) is the graphene layer and (SC) is a superconductor. In order to avoid the high technologically challenging aspects of the GR/SC substrate we opted for n-doped Si substrate, which may have similar anti-reflective characteristics. The resulting device structure was then Si/I/M triple layer. In some cases we have investigated quadruple layer devices Si/I/M1/M2 made up of several metal nano-films or a combination of metal/semiconductor nano-films. All metal nano-films were grown by e-beam deposition on thick n-doped Si substrate, covered by a thick oxide layer made of SiO_2_ or sapphire, Al_2_O_3_. The SiO_2_ layer had thicknesses of 540 nm and 430 nm, while the thinner Al_2_O_3_ layer was 60 nm. The nm-size metallic film had a varied surface morphology and thickness. Schematic diagram of the experimental setup is shown in Figure 1. All studied devices show strong effects towards the perfect refractive index matching or 100% anti-reflection and transmission arising at some optimal thickness of the metal nano-film. In excellent agreement with theoretical predictions, the experimental data shows that the reflectivity vanishes only at specific values of the metal nano-film thickness which depends on the type of material used. The phenomenon is seen to occur practically for all types of metal nano-films used, in certain cases extending over a select frequency range. In essence, the effect is related to a destructive interference happening inside the oxide layer, which is enhanced by the presence of the ultra-thin, nm-size metallic film on its surface.

To illustrate that the phenomenology and the approach is quite general we have initially investigated devices with Pt nano-films—Si/SiO_2_/Pt. In this case the Pt nano-films were deposited onto a SiO_2_ oxide on an n-doped Si substrate. The thickness of SiO_2_ layer was 430 nm. The thickness of the Si substrate was 1 mm. The Pt film had thicknesses 2 nm and 5 nm. The reflectivity dependence on incident light wavelength in the Pt nano-film demonstrate a double minima, occurring near wavelengths of λ_1_ = 0.5 µm and λ_2_ = 0.85 µm, respectively (see, Figure 5a). The colours correspond to Pt thicknesses of 0 nm (blue), 2 nm (orange) and 5 nm (green). The solid curves are the results of the theoretical models based on a transfer matrix method, where the refractive index of the Pt film has been chosen to fit the data. It is particularly evident that for Pt nano-film with thickness of 2 nm the reflectivity approaches 0.09% and the transmission is 99.91% near 100% anti-reflection limit at wavelengths around 840 nm (see values in Table 2). Figure 5b demonstrates the dependence of the effective refractive index n_eff_ on the thickness of the Pt nano-film. Notably, when the Pt nano-film thickness reaches 2 nm the effective refractive index n_eff_ has a real part close to 1 and a negligibly small extinction coefficient over a wide wavelength range (see Figure 5b solid orange curve). In essence for this particular Pt thickness the *n*_eff_ of the triple layer device approaches the 100% anti-reflection limit given by the condition *N*_eff_ = 1 + 0 *i* (see Figure 5b blue dotted line). This is seen to happen at λ ~ 0.4–0.6 µm and λ ~ 0.8–0.9 µm which coincides precisely with where experimentally the reflectivity minima (and transmission maxima) are observed (see Figure 5a).

To complete our understanding and test the rigour and the extent of our phenomenological model, we have investigated its predictions for the THz frequency range. We have used the model to calculate the reflectivity profiles for a triple layer material made up of Nb nano-films deposited on sapphire (Al_2_O_3_) oxide on a Si substrate. The thickness of the Al_2_O_3_ was chosen as 80 μm to complement the THz wave range. The behaviour of the reflectivity of the Nb triple layer structure was studied at incident THz wavelength of λ=100 μm. Specifically, we have imposed the condition for perfect impedance matching, or *N*_eff_ = 1 + 0 *i*, and calculated the reflectivity profiles for different thicknesses of the Nb nano-films *d*_1_ (see Figure 6). The 100% antireflection regions are indicated in dark purple/black. Our model confirms that 100% antireflection at THz frequencies can occur for a specific choice of Nb nano-film thickness. In all cases the calculated values of the refractive index n fall within the range n ~ 30–50 at THz frequencies, which is viable for the materials used and is in excellent agreement with published literature [43,44]. In particular, instead of Nb, we may use conducting ZnO nano-films. The results for the THz refractive index and conductivity of the ZnO films obtained in the ref. [43] fall just inside the range of our prediction for perfect impedance matching (see Figure 6a). It is plausible that ZnO could be also utilised as part of our triple layer structures, for example, as Si/Al_2_O_3_/ZnO meta-surface, where the thickness of the sapphire is 80 μm and the thickness of ZnO is 2 nm, respectively.

## 5. Discussion

However, the most efficient anti-reflective properties were observed in quadruple layer devices with nano-films made up of two types of materials Ge and Sn—Si/Al_2_O_3_/Ge/Sn. In these hybrid meta-surfaces, the metal nano-film is replaced by a combination of Ge and Sn layers of equal thicknesses deposited onto a 60-nm oxide layer of Al_2_O_3_. The Ge layer is believed to act as a wetting buffer layer needed to maintain a homogeneous morphology of the Sn nano-film and ensuring its high quality. The 100% anti-reflection and transmission was observed for meta-surfaces with Ge and Sn nano-film thicknesses equal to 1 nm (see Table 2). Table 2 summarises the parameters for 100% anti-reflection and transmission observed in the studied triple and quadruple layer structures. Our results show that the discovered phenomenon of 100% anti-reflection in metal nano-film capped meta-surfaces appears to be very general, and may be achieved practically for any frequency range by appropriately choosing the metal and oxide combinations.

Note that the effect of perfect refractive index matching or 100% anti-reflection may also be achieved by varying the angle of the incident light and its polarisation. The behaviour of the reflectivity on the incident angle of light with s-polarisation for Si/Al_2_O_3_/Ge/Sn and Si/SiO_2_/Nb meta-surfaces is shown in Figure 7. In this case the thicknesses of the nano-films have been chosen to correspond to the 100% anti-reflection condition at normal light incidence—1 nm thick Ge/Sn and 10 nm thick Nb. The results for the Si/Al_2_O_3_/Ge/Sn meta-surface are presented in Figure 7a. Here, the thickness of the Al_2_O_3_ was 60 nm. The reflectivity reaches a minimum of 0.0002 at an incident angle of 40° (Figure 7a (blue line)), corresponding to a transmission of 99.98%. Interestingly, the wavelength of this minimum ~414 nm is slightly lower than for light at normal angle of incidence. The reflectivity for the Si/SiO_2_/Nb is shown in Figure 7b. The thickness of the SiO_2_ is 540 nm. The reflectivity reaches a minimum of 0.0003 at an incident angle of 30° (Figure 7b (black line)), corresponding to a transmission of 99.97%. Notably, the wavelength of this minimum 850 nm is substantially higher than for light at normal angle of incidence. In this particular case also a higher transmission value of 99.97% is achieved compared to normal light incidence. It is likely that for the Nb meta-surface multiple minima are seen because there are many more interference peaks due to the thicker SiO_2_ layer. The lowest reflectivity value of 0.0002–0.0003 for both types of meta-surfaces is obtained at an angle of 40° in Si/Al_2_O_3_/Ge/Sn and at an angle of 30° in Si/SiO_2_/Nb films.

Furthermore, our phenomenological approach for predicting 100% anti-reflection has been extended to THz frequencies. We have demonstrated that perfect impedance matching can arise in trilayer materials made up of metallic nano-films and oxide deposited on a Si substrate. Our calculations show viable refractive indices in the range of n ~ 30–50 for the THz range. These values of n are very similar to what has been reported in the literature. For example, it was found that the refractive index of the ZnO film over the THz range of frequencies, n ~ 30–90 is much larger than that of the glass substrate, whereby the extinction coefficient associated with the conductivity of the film is almost frequency independent [43]. Additionally, it was shown that the intensity of broadband terahertz pulses can be suppressed to below 1% with nanostructured Au films on Si substrate [44]. The result has been supported by finite element method simulations illustrating impedance matching at Au film thicknesses of about 7.5 nm. The superior performance of such gold clustered films is due to carrier localisation enhanced by back-scattering from the nanostructures. Both examples, the clustered Au and ZnO films demonstrate nearly frequency-independent optical conductivity in the THz range. These observations show striking agreement with the predictions of our phenomenological model, where 100% anti-reflection is obtained for Nb nano-films with thicknesses 2–8 nm, with a large refractive index n~30 at THz frequencies. Thus, our work highlights the possibility to perfectly match wave impedances between substrates and air over a broadband width of frequencies by choosing the optimal metal nano-film thickness in a conjunction with an insulating layer.

As an aside, in recent works, it was noted that non-resonant anti-reflection coatings in a material with sub-wavelength thickness can be realised using thin films where either the permittivity or permeability are dominated by imaginary parts [45]. Such lossy or gain coating can reduce or enhance the transmitted waves, respectively. The effect depends on light polarisation and incidence angle. When light has a transverse magnetic polarisation, the same coating may switch between gain and loss performances at Brewster angle. The findings have been verified for microwaves. The extension of a similar approach to the fabrication of anti-reflection coatings operating in the THz range would signify the next important leap in THz technology. We propose that the newly suggested method for creating anti-reflection meta-surfaces using metallic nano-film and oxide combinations may offer one such possibility.

In conventional anti-reflective coatings, for example those used industrially in solar panels and cameras, normally an insulating SiO_2_ plate is covered by a graded SiO_2_/TiO_2_ film that acts as a destructive quarter wavelength interference layer. The other frequent methods for producing anti-reflection coatings utilise complex surface structures similar to moth-eye nano-tubules [46] or a combination of multilayers [47,48]. Such surfaces are often determined empirically to achieve the most optimal anti-reflection performance. The inclusion of anti-reflective coatings improves the efficiency of existing optoelectronic devices by minimizing light-loss due to reflection and removing glint. As a rule, in telescopes lenses multilayer coatings are preferred [47,48] to eliminate stray light from stellar observables. The moth-eye nanostructured anti-reflection coatings are often used in light-emitting diodes [49], displays [50], photovoltaic solar cells [51] and micro solar sensors [52]. The described technologies where the reflectivity is reduced through optical destructive interference at best give transmission rates of about 90% [53].

Conventional anti-reflection approaches [51,52,53] are not entirely suitable to the lower frequency THz and the microwave regimes. The issues arise because such coatings must be greater than the half-wavelength of incident light [54,55], which limits the size and range of the existing microwave and THz devices [56]. In particular, the same problem occurs in anti-reflection technology for microwave and THz transmission lines where, so far, various impedance matching transformers have been employed [57,58].

In the present paper, in our anti-reflective coating we have added an extra nano-meter thick metal layer of material compared to the most common (and simple) designs for anti-reflective films used for many years in diverse applications, e.g., in solar panels and cameras. We have discovered that due to this extra layer the light reflectivity drops to zero and we were able to achieve perfect impedance matching for several studied surfaces. We propose that the method can be adopted to any existing designs, where the quality of anti-reflection can be increased to nearly 100% by depositing a very thin nm size metallic layer on top of the fabricated structures. The additional nm-metal coating does not practically increase the size (and cost) of the devices or affect its optical paths, but enables perfect impedance matching to occur widely across the electromagnetic spectrum, including in the microwave and THz ranges. These results may signal a new era in optoelectronic devices and open up novel avenues for diverse applications.

## 6. Conclusions

In summary, we have found a novel method, utilising nm-size metal films in combination with oxides, to create meta- surfaces with perfect refractive index matching or 100% anti-reflection for any frequency range, including the THz waves. The newly discovered meta-surfaces can be explained by introducing an effective refractive index associated with the metal-oxide-n-doped Si, triple layer structure. In order to obtain the effective optical constants of this triple layer system, we have derived its transfer matrix description, see Equation (3). Through rigorous analysis, our modelling demonstrates that because of the inclusion of the metal nano-film on its surface the effective refractive index of the triple layer system can be adjusted to match that of air. In other words, particular tuning of the thickness of the metal nano-film coating may give rise to 100% anti-reflection from the entire system. These predictions are experimentally confirmed in several fabricated meta-surfaces over a wide range of frequencies. Particular success has been obtained in meta-surfaces with quadruple layers Si/Al_2_O_3_/Ge/Sn, where never before achieved reflectivities well below 0.01% or transmission rates above 99.99% have been observed. This is more than an order of magnitude greater compared to existing state-of-the-art technologies. The phenomenon may be linked to a hybridisation effect between the metal nano-film and the underlying oxide, resulting in a lower refractive index than the bulk and negative permittivities. Our phenomenological approach together with the proposed simple and robust methodology opens up avenues to achieve the perfect 100% anti-reflective coatings practically for any desired frequency range. This work and its findings should facilitate new developments and advances in all sectors and industries working to minimise light reflection and THz wave losses for cultural and economic progress.

## Figures and Tables

**Figure 1 materials-13-05417-f001:**
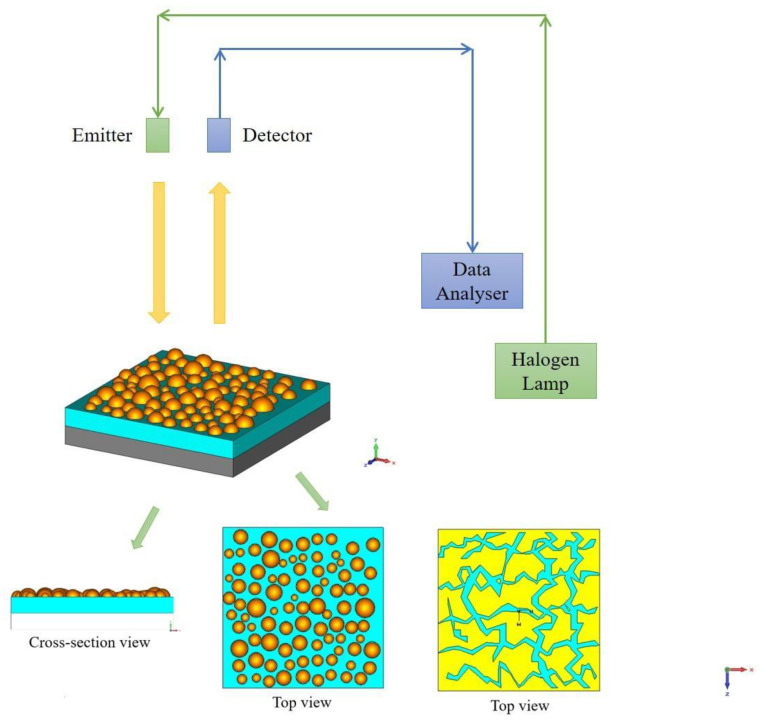
Schematic experimental setup for the reflectivity experiment. The tested triple layer samples consist of: A thick n-doped Si substrate (shown in grey); an insulating SiO_2_ or Al_2_O_3_ oxide layer with thickness of 540 nm and 60 nm, respectively (shown in cyan); and a nm-sized metal film (shown in yellow). The role of the metal nano-film on the surface of the oxide is to create effective refractive index matching between the air and the system. The morphology of the metal nano-film varies depending on the film thickness. In the thinnest Sn nano-films, the Sn metal was deposited onto a homogeneous and continuous Ge buffer layer (1 nm thick). The resulting film consists of separate droplets (see Top view left). In the thickest Nb films the morphology resembles more interconnected islands (see Top view right). The incident light is directed both perpendicular to the flat film surface and at some angles. The wavelength of the incident light is varied from near ultra-violet (425 nm) to near infrared (920 nm), completely covering all visible light frequencies.

**Figure 2 materials-13-05417-f002:**
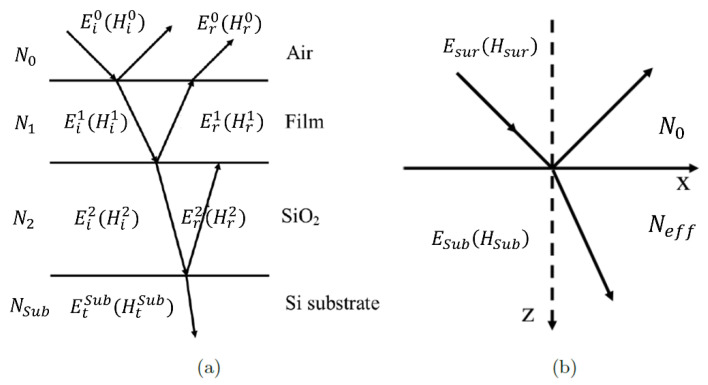
The schematic diagram of the transmitted and reflected waves from (**a**) triple layer (**b**) single interface structures. The reflectivity at each interface is described using Fresnel equation and depends on the optical constants of the materials at each interface. The reflectivity of the triple layer structure is determined by the interference between the reflected waves from all interfaces and the transmitted waves through all layers.

**Figure 3 materials-13-05417-f003:**
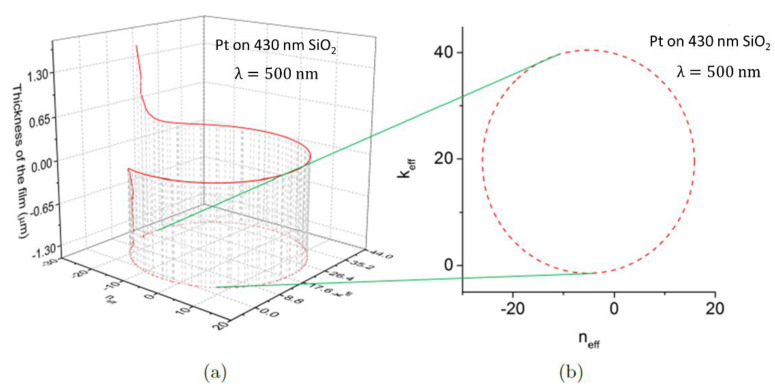
(**a**) The helical dependence of the metal nano-film thickness, *d*_1_ on the trilayer optical constant, *N*_eff_ = *n*_eff_ + i*k*_eff_ in the (*d*_1_, *n*_eff_, *k*_eff_) space. The film thickness *d*_1_ is calculated using Equation (4). By changing the nano-film thickness the optical constant *N*_eff_ will follow the helical trajectory shown in red. This illustrates that the optical constant of the film is largely determined by the film thickness. (**b**) The projection of *d*_1_ on the (*n_eff_, k_eff_*) plane is a circle. The trajectory has been calculated for ultra-thin Pt film, which *N*_1_ = 3.20 − 4.3 *i*, deposited on 430 nm of SiO_2_ on Si.

**Figure 4 materials-13-05417-f004:**
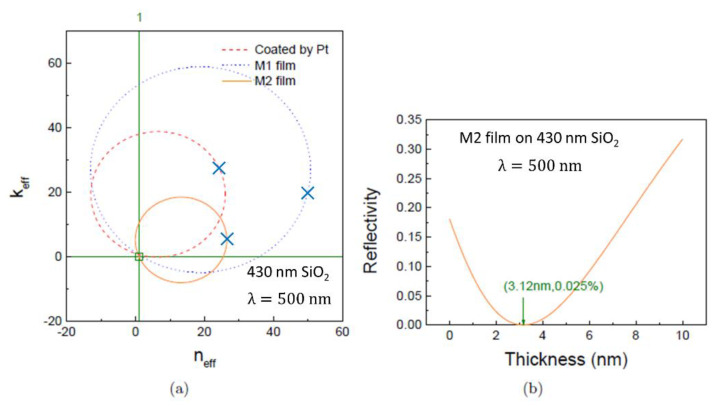
(**a**) The examples of theoretical modelling where the effective optical constant, *N*_eff_, can achieve the value 1 + 0 *i*, i.e., 100% anti-reflection. They are obtained for a triple layer structure, consisting of a thin metal nano-film, an insulating oxide layer such as SiO_2_ and a substrate such as n-doped Si. The circles are the 2D projections of the *d*_1_ trajectories onto the (*n*_eff_, *k*_eff_) plane as explained in Figure 3. Different circles of *N*_eff_ correspond to different values of the optical constants, *N*_1_. The red (dashed) projection is obtained for a Pt film with *N*_1_ = 3.20 − 4.3 *i*. The blue (dashed) and orange (solid) projections represent artificial metals M1 and M2 with *N*_1_ = 3.2 − 6 *i* and *N*_1_ = 0.86 − 4.3 *i*, respectively. All *N*_1_ values are taken from Table 1. For example, the orange circle associated with the film M2 crosses the horizontal axis at position (1,0), *N_eff_* = 1 + 0 *i*, which implies 100% anti-reflection or zero reflection. The (*n_eff_, k_eff_*) values corresponding to the absence of the metallic film (*d*_1_ = 0 nm) are marked as blue crosses for Pt, M1 and M2 films. (**b**) The dependence of the reflectivity on the M2 film thickness *d*_1_ is shown. The reflectivity reaches a minimum at a critical thickness value *d*_1_ = 3.12 nm. At this point the minimal reflectivity is equal to 2.5 10^−4^ or 0.025%. Conversely, the transmission at the minimum is 99.975%.

**Figure 5 materials-13-05417-f005:**
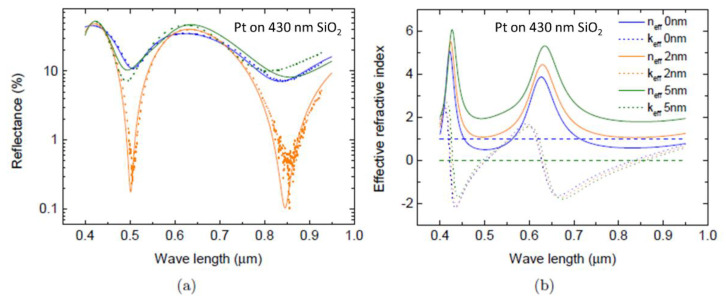
(**a**) The reflectivity wavelength dependence of the Pt nano-film triple layer systems. The Pt nano-film thicknesses correspond to 0 nm (blue), 2 nm (orange) and 5 nm (green). The light wavelength λ is given in µm. The solid curves are the results of the theoretical models based on a transfer matrix method. The 100% anti-reflection and transmission limit is reached in the 2 nm Pt nano-films with a minimum reflectivity value of 0.09% around 0.85 µm. (**b**) The effective complex refractive index calculated at normal incidence for Pt triple layers at different Pt thicknesses. The blue, orange and green lines represent Pt films with thicknesses 0, 2 and 5 nm respectively. The solid and dashed lines are *n*_eff_ and *k*_eff_ respectively.

**Figure 6 materials-13-05417-f006:**
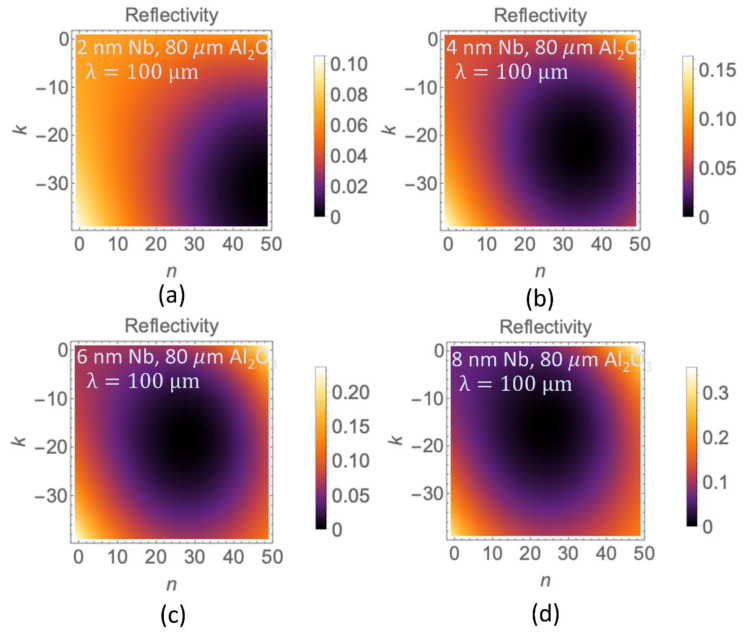
The dependence of the reflectivity on optical parameters *n* and *k* calculated at film thickness *d*_1_ and light wavelength λ=100 μm corresponding to perfect impedance matching condition. Black/dark purple regions correspond to 100% anti-reflection. The Nb nano-film thicknesses were (**a**) 2 nm, (**b**) 4 nm, (**c**) 6 nm and (**d**) 8 nm.

**Figure 7 materials-13-05417-f007:**
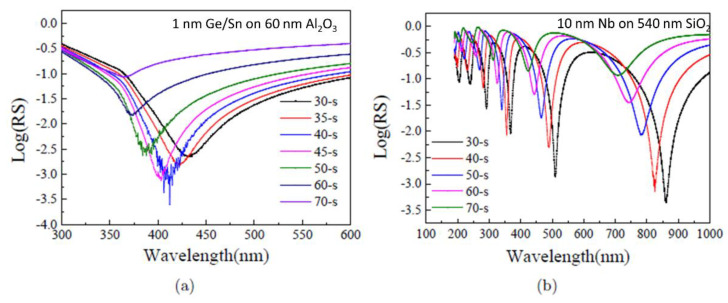
The dependence of the reflectivity measured at different angles of light incidence to the plane in triple and quadruple layer meta-surfaces. The n-doped Si was used as a substrate in all experiments. The incident light had s-polarisation. The light angle is measured from the normal to the meta-surface plane. The logarithm of reflectivity is shown, reflectivity RS is given as absolute value. The value of the incident angle is shown in different colours in the figure panel. (**a**) Quadruple layer meta-surface made of Si/Al_2_O_3_/Ge/Sn. (**b**) Triple layer meta-surface made of Si/SiO_2_/Nb. Note, that the lowest reflectivity value of 0.0002–0.0003 for both types of meta-surfaces is obtained at an angle of 40° (blue line) in Si/Al_2_O_3_/Ge/Sn and at an angle of 30° (black line) in Si/SiO_2_/Nb films. For Nb meta-surfaces multiple minima in reflectivity are observed, which may be related to the thicker oxide layer in this structure.

**Table 1 materials-13-05417-t001:** The optical constants for different compositional materials used in triple layer systems with 100% anti-reflection. M1 and M2 refer to artificial materials. M2 has optical constants similar to many metals. The values were obtained for incident wavelength of λ = 500 nm.

Material	Refractive Index	Thickness
Si	4.3 − 0.049 *i*	>10 µm
SiO_2_	1.55	430 nm
Pt	3.2 − 4.3 *i*	<10 nm
M1	3.2 − 6 *i*	<10 nm
M2	0.86 − 4.3 *i*	<10 nm

**Table 2 materials-13-05417-t002:** Experimentally determined parameters of meta-surfaces where perfect impedance matching and 100% anti-reflection has been found. The meta-surfaces were triple layers, consisting of a substrate (Si), an oxide (I) and a metal (M) nano-film layer. Only cases corresponding to transmission greater than 99% and reflectivity less than 1% have been selected. For all samples, the minimal value of the reflectivity R, maximal transmission value T and the associated wavelength λ were recorded. In the case of Ge/Sn films the thickness of the metal Sn film has been carefully tuned to achieve 100% anti-reflection. It should be possible to repeat the same process for other nano-film combinations. All numbers in table are measured in nm and the reflectivity and transmission are given in %. In certain cases combinations of several metal nano-films where tested, and the thickness ratio indicated as appropriate.

Metal M	Oxide Layer I	*d*_1_, nm	h, nm	R_1_ (%)	T_1_ (%)	λ_1_ (nm)	R_2_ (%)	T_2_ (%)	λ_2_ (nm)
Sn/Au	SiO_2_	2/3	430	0.098	99.902	523.4	0.766	99.234	918.8
Sn/Au	SiO_2_	2/3	540	0.540	99.46	451.1	0.508	99.492	643.1
Bi	SiO_2_	6	540	0.624	99.376	452.3	0.078	99.922	637.4
Bi	SiO_2_	7	540	0.150	99.85	453.0	0.640	99.36	638.0
Nb	SiO_2_	10	540	0.056	99.944	458.8	0.112	99.888	646.0
Ag	SiO_2_	5	540	4.090	95.91	443.5	0.632	99.368	659.3
Pt	SiO_2_	2	430	0.225	99.78	504.6	0.095	99.905	842.7
Ge/Sn	Al_2_O_3_	1/1	60	0.000	100	434–437	0.35	99.65	460.0

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
