# Peer review of "Perfect Impedance Matching with Meta-Surfaces Made of Ultra-Thin Metal Films: A Phenomenological Approach to the Ideal THz Sensors"

_materials, 2020, doi:10.3390/ma13235417_

Round 1

Reviewer 1 Report

1. The authors were not very careful in preparing the article and the main idea, the independence of the method with frequency, does not seem to make sense (see point 2 below). From my point of view the article has very little value if the method is not independent of frequency as claimed. In that case, the authors should realistically compare their technique with other techniques in the literature and show evidence that their technique is better.

2. There are figures without labels on both axes. The numbering of sections and figures was not taken care of. There are two sections 3: Methods and Results. Thus, Methods should be section 3, Results should be section 4, Discussion should be section 5 and Conclusions should be section 6. There are three Figures named Fig. 5, one on lines 240-255. Another on lines 256-282. Another on lines 314-336.

3. The presented examples do not work as stated. Moreover, the examples are not correctly formulated, important data is not given: wavelength / frequency. The authors have to make this very clear.

Proof: in lines 190-204 the authors give an example for material M2. The parameters for this example are:

# Film (M2)
N1 = 0.86 - 4.3i
d1 = 3.12 nm

# SiO2
N2 = 1.55
d2 = 430 nm

# Si
N3 = 4.3-0.049i
d3 = 15 nm # they did not gave this value, just stated >10um in table 1
# the value is irrelevant since, although not explained in the text, the Si layer
# is not terminated and as can be seen in formulas (3) and (4)
# the values of N3 and d3 are not used. I agree with this.

Using these parameters we can calculate Neff with formula (3).We should get Neff = 1, according to the authors because of impedance matching. But we need delta_1 and delta_2:

delta_1 = 2 * pi * N1 * d1 / lambda (see line 149)
delta_2 = 2 * pi * N1 * d2 / lambda (see line 149)

So we need the wavenumber --> lambda to calculate Neff. The authors do not provide this value and  claim a formulation independent of frequency (line 367):

"Our results show that the discovered phenomenon of 100%
anti-reflection in metal nano-film capped meta-surfaces appears to be very general, and may be
achieved practically with any metal and for any frequency range."

or (line 389):

"In summary, we have found a novel method, utilising nm-size metal films in combination with
oxides, to create meta- surfaces with perfect refractive index matching or 100% anti-reflection for any
frequency range, including the THz waves."

Let's take a random frequency in the THz gap: lambda = 1 mm.

In this case we get: Neff = 4.3 - 0.09 i, not Neff as claimed.

Another example: lambda = 100 um --> Neff = 4.239684 - 0.483141 i, not Neff as claimed. 

authors must clarify this.

4. In line 144, position M_22 of the matrix should be cos (delta_m), not delta_m.

5. In addition, Nm is said to be the refractive index (see line 148), which is not true if we assume that E and H are the electric and magnetic fields. In fact, it is said in the text that E and H are the electric and magnetic fields (see line 140). Therefore, N_m should be the admittance of layer m, not the refractive index as stated. The document needs to be corrected and clarified because the authors are using a normalized electric field (the normalization constant is the free space impedance). So the authors should explicitly refer this normalizatoin. Otherwise it is strange to read, for example, in line 150, that the division of a magnetic field by an electric field gives an index of refraction. The important issue is that the formulas that follow (1) refer to this transformed fields and the authors do not make this clear.

6. In line 149, dm appears and its meaning was not mentioned in the text. The authors have to fix this.

The relationship between Esur, Hsur and the fields in the figure is not stated explicitly. The same for Esub and Hsub. I suggest adding this.

7. In line 141, H_para is mentioned but no clarification about its meaning was given. It seems to be irrelevant since it is not necessary to follow the article. My suggestion is that it be removed.

8. In fig.2a in place of N3 should be Nsub. N3 is never referred in the text, Nsub is always used for the refractive index of the Si substrate. See for example in formulas (2) and (3).

9. In fig 2b, N1 is also never used. Neff should be in the place of N1 as mentioned in lines 137-139 as it makes more sense in the context of the article.

10. In line 204, where it is stated "as calculated by Eq. 3" i think it is "as caculated by Eq. 4".

11. Fig. 5 has no x axis and no y axis.

Author Response

Referee 1:

We would like to thank the referee for his valuable insight and comments, suggestions and constructive guidance. Following his suggestions, we have revised and rewritten a large part of the paper. All the inserted modifications and corrections are highlighted in blue in the text. Furthermore, we provide a point by point address to specific queries raised by the referee. Please see, details below:

  1. The authors were not very careful in preparing the article and the main idea, the independence of the method with frequency, does not seem to make sense (see point 2 below). From my point of view the article has very little value if the method is not independent of frequency as claimed. In that case, the authors should realistically compare their technique with other techniques in the literature and show evidence that their technique is better.

We thank the referee for raising this important point and we have rewritten large parts of the paper accordingly. We have expanded our paper and substantially amended the Introduction, which now includes achievements and progresses in the THz field. We have included a section on reflectivity in the THz range. Specifically, we have used our phenomenological approach to simulate the reflectivity behaviour of our triple layer materials in the THz range. The results have been presented in a new Figure 7 together with an accompanying explanation in the text. The predictions of refractive indices by our model are in good agreement with published literature. This has been discussed in detail in the Results and Discussion sections.

We believe that our proposed methodology for creating anti-reflective coatings by combining metal nano-films and oxides has almost limitless potential across the visible light spectrum all the way to THz waves. Our experimental and theoretical results show that the appropriate choices of metal nano-film and  oxide thicknesses can target anti-reflection almost at any frequency range. The performance of the triple layers can be tuned to perfect 100% anti-reflection by covering the oxide with a metallic nano-film. Our results are seen to be true for the visible light and the infra-red parts of the spectrum. Our theoretical simulation confirm that it will also be present in THz ranges. The refractive indices we are predicting are well in agreement with published literature of what has been experimentally observed in the THz range. We have inserted appropriate references of these observations. Furthermore, we have achieved anti-reflectivity that is over an order of magnitude better than what is available with current state-of-the-art technologies that can obtain reflectivities as low as 0.1%. Our triple layer samples can attain reflectivities well below 0.01% that is the limit of the experimental apparatus.

  1. There are figures without labels on both axes. The numbering of sections and figures was not taken care of. There are two sections 3: Methods and Results. Thus, Methods should be section 3, Results should be section 4, Discussion should be section 5 and Conclusions should be section 6. There are three Figures named Fig. 5, one on lines 240-255. Another on lines 256-282. Another on lines 314-336.

We thank the referee for noticing. We have corrected all the numberings of sections and figures.

  1. The presented examples do not work as stated. Moreover, the examples are not correctly formulated, important data is not given: wavelength / frequency. The authors have to make this very clear.

Proof: in lines 190-204 the authors give an example for material M2. The parameters for this example are:

# Film (M2)
N1 = 0.86 - 4.3i
d1 = 3.12 nm

# SiO2
N2 = 1.55
d2 = 430 nm

# Si
N3 = 4.3-0.049i
d3 = 15 nm # they did not gave this value, just stated >10um in table 1
# the value is irrelevant since, although not explained in the text, the Si layer
# is not terminated and as can be seen in formulas (3) and (4)
# the values of N3 and d3 are not used. I agree with this.

Using these parameters we can calculate Neff with formula (3).We should get Neff = 1, according to the authors because of impedance matching. But we need delta_1 and delta_2:

delta_1 = 2 * pi * N1 * d1 / lambda (see line 149)
delta_2 = 2 * pi * N1 * d2 / lambda (see line 149)

So we need the wavenumber --> lambda to calculate Neff. The authors do not provide this value and  claim a formulation independent of frequency (line 367):

"Our results show that the discovered phenomenon of 100%
anti-reflection in metal nano-film capped meta-surfaces appears to be very general, and may be
achieved practically with any metal and for any frequency range."

or (line 389):

"In summary, we have found a novel method, utilising nm-size metal films in combination with
oxides, to create meta- surfaces with perfect refractive index matching or 100% anti-reflection for any
frequency range, including the THz waves."

Let's take a random frequency in the THz gap: lambda = 1 mm.

In this case we get: Neff = 4.3 - 0.09 i, not Neff as claimed.

Another example: lambda = 100 um --> Neff = 4.239684 - 0.483141 i, not Neff as claimed. 

authors must clarify this.

We apologize for the confusion. We thank the referee for raising this important point and agree with his given examples. As stated, the reflectivity of these materials does depend on frequency. We have clarified this in the text and added frequencies as necessary to all simulations and the results in the figures (Figures 3, 4, and 7) as well as tables (Table 1). The novelty of our novel approach allows to target any particular frequency band by choosing the appropriate combinations of oxide and metal nano-film, and by varying their respective thicknesses. As we have demonstrated experimentally and through theoretical simulations, minute (less than 1 nm) variations in the metal nano-film thicknesses influence the reflectivity dramatically and can at specific thickness values produce 100% anti-reflection. The wavelength position of the reflectivity minimum is related mostly to the thickness of the oxide. Our results demonstrate this clearly. For example, 100% anti-reflection occurs at lambda=400nm when the oxide is 60micrometers, at lambda=900nm when the oxide is 400 – 500nm, and at lambda=100micrometers when the oxide is 80micrometers. We have amended this in the paper text. Nevertheless, our results demonstrate that the proposed method to achieve100% anti-reflectivity is universal and can apply for all frequencies of the visible light through to the terahertz waves.

  1. In line 144, position M_22 of the matrix should be cos (delta_m), not delta_m.

We have rectified this typo, and generally improved the presentation of the formulas in the paper.

  1. In addition, Nm is said to be the refractive index (see line 148), which is not true if we assume that E and H are the electric and magnetic fields. In fact, it is said in the text that E and H are the electric and magnetic fields (see line 140). Therefore, N_m should be the admittance of layer m, not the refractive index as stated. The document needs to be corrected and clarified because the authors are using a normalized electric field (the normalization constant is the free space impedance). So the authors should explicitly refer this normalizatoin. Otherwise it is strange to read, for example, in line 150, that the division of a magnetic field by an electric field gives an index of refraction. The important issue is that the formulas that follow (1) refer to this transformed fields and the authors do not make this clear.

We thank the referee for highlighting the importance of normalized fields and units. We have added the proper normalisation constant – which is one over the intrinsic impedance of vacuum gamma=1/Z0. Please, see lines 205 – 213 of the paper. (Z0 - the impedance of free vacuum was also introduced, in line 89.) In this formulation N_m now properly refers to the refractive index, and not admittance as was in the previous version of the paper.

  1. In line 149, dm appears and its meaning was not mentioned in the text. The authors have to fix this.

The relationship between Esur, Hsur and the fields in the figure is not stated explicitly. The same for Esub and Hsub. I suggest adding this.

We thank the referee for noticing. In the text we have clarified “m” to be the index of the layers inside the multilayer structure, whereby “dm” is the phase shift associated with each layer. Please, see lines 226 – 230. As the referee suggested, we have also added and clarified the relationship between Esur, Hsur and Esub and Hsub as indicated in Figure 2. Please, see line 200 – 201 as well as the new notations in Figure 2b.   

  1. In line 141, H_para is mentioned but no clarification about its meaning was given. It seems to be irrelevant since it is not necessary to follow the article. My suggestion is that it be removed.

We agree with the referee that H_para is unnecessary and remove it from the manuscript.

  1. In fig.2a in place of N3 should be Nsub. N3 is never referred in the text, Nsub is always used for the refractive index of the Si substrate. See for example in formulas (2) and (3).

We thank the referee for noticing, we have corrected Figure 2a as suggested by replacing N3 with NSub.

  1. In fig 2b, N1 is also never used. Neff should be in the place of N1 as mentioned in lines 137-139 as it makes more sense in the context of the article.

We thank the referee for noticing, we have corrected Figure 2b as suggested by replacing N1 with Neff.

  1. In line 204, where it is stated "as calculated by Eq. 3" i think it is "as caculated by Eq. 4".

We thank the referee for noticing, we have corrected the typo.

  1. Fig. 5 has no x axis and no y axis.

We agree with the referee that Figure 5 has little information to contribute to the paper, and have hence removed it from the manuscript, replacing it with Figure 7, containing simulation reflectivity results for the THz regime.

Reviewer 2 Report

Author investigate new method based on meta-surfaces made of ultra-thin metal films to enhance the reflectivity up to 100% and transmittance over a wide range of wavelength. I believe that the subject is of great interest for the community and especially for terahertz technology. However, the paper lack of results and discussion. Below are my comments to improve the quality of the paper.

1- The author claim that these materials are suitable for terahertz technology. However no experimental (or theoretical) were presented in the manuscript. Author investigate the visible up to the infrared range. 

2- They presented structure using 2d material like graphene but no graphene was studied in the paper. 

3- Figure 3 shows the metal-nanofilm thickness dependence in (neff,keff) space but keff was not defined in the text. It should be the imaginary part of neff.

4- Figure 4 shows the modelling results of diferente metal nano films (M1, M2, coating with Pt). It will be good if the author include the reference one, the structure without the metal film.

5- Figure 5 was included but no word in the text

6- Results shown in figures 6 and 7 does not present the axis x and y (no label and no values??)

7- The authors claim that there method could reach 100% reflectivity, however, the demonstrate a value around 99.97%. However, and within this range, the actual technology already provide 99.99% as stated by the authors. What is the new/interest of their technology in this case.

Minro comments:

1- Caption of figures are too long and better include the description in the figure and in the manuscript.

2- Figures 6 and 7 are shown as 5

3- The figures overall are poorly presented and should be improved by including a correct description in the figures and not in the captions. Legend are mixed with the plots,...

Author Response

Referee 2:

We would like to thank the referee for his valuable insight and comments, suggestions and constructive guidance. Following his suggestions, we have revised and rewritten a large part of the paper. All the inserted modifications and corrections are highlighted in blue in the text. Furthermore, we provide a point by point address to specific queries raised by the referee. Please see, details below:

1- The author claim that these materials are suitable for terahertz technology. However no experimental (or theoretical) were presented in the manuscript. Author investigate the visible up to the infrared range. 

We thank the referee for raising this important point. We have expanded our paper, and included a section on reflectivity in the THz range. The Introduction section now includes achievements and progresses in the THz field. We have used our phenomenological approach to simulate the reflectivity behaviour of our triple layer materials in the THz range. The results have been presented in a new Figure 7 together with an accompanying explanation in the text. Our model predictions of refractive indices of metal nano-films and oxide layers, that provide a perfect impedance matching, are in good agreement with published literature. This has been discussed in detail in the Results and Discussion sections.

2- They presented structure using 2d material like graphene but no graphene was studied in the paper. 

We thank the referee for the perceptive comment. We have modified the Introduction to include a more wider review of existing anti-reflective material technologies, not limiting to 2D systems.

3- Figure 3 shows the metal-nanofilm thickness dependence in (neff,keff) space but keff was not defined in the text. It should be the imaginary part of neff.

We agree with the referee that Figure 3 is not explained properly. We have added clarification on Figure 3 in the text, explaining the symbols and the relevance.

4- Figure 4 shows the modelling results of diferente metal nano films (M1, M2, coating with Pt). It will be good if the author include the reference one, the structure without the metal film.

We thank the referee for the insightful suggestion. The reference optical constants for materials with no metallic coating have been added to Figure 4 as blue crosses.

5- Figure 5 was included but no word in the text

We agree with the referee that Figure 5 has little information to contribute to the paper, and have hence removed it from the manuscript, replacing it with Figure 7, containing simulation reflectivity results for the THz regime.

6- Results shown in figures 6 and 7 does not present the axis x and y (no label and no values??)

We have corrected all figures. The figures have now appropriate axis, labels, legends. The relevant information is included in directly in the plot.

7- The authors claim that there method could reach 100% reflectivity, however, the demonstrate a value around 99.97%. However, and within this range, the actual technology already provide 99.99% as stated by the authors. What is the new/interest of their technology in this case.

We have clarified in the text that existing technologies provide transmission rates up to 99.9%. In the present paper we show materials that demonstrate transmission rates over 99.99%. This performance is an order of magnitude or more greater than what can be achieved with establish anti-reflection coatings.

Minro comments:

1- Caption of figures are too long and better include the description in the figure and in the manuscript.

We have shortened the captions of all figures, as the referee has suggested. The information has been included in the manuscript text and directly in the plot.

2- Figures 6 and 7 are shown as 5

We have corrected the figure numbering.

3- The figures overall are poorly presented and should be improved by including a correct description in the figures and not in the captions. Legend are mixed with the plots,...

We have corrected all figures. All figures now have the appropriate labels, legends and scales. In addition, we have added relevant content to the plots themselves.

Round 2

Reviewer 1 Report

The paper was corrected according to the recommendations. My point of view became much clearer with the authors' response.
On the positive side, the authors did not limit themselves to filing the edges of what they left poorly explained in the first version since they added new figures that seem to improve the
quality of the work.

The previous document was clearly not finished or ready for review, the authors didn't give their best
to ensure that the work would make the best impression right from the start. In particular, the work was not at all clear in showing the frequencies in which
the study was done and how it behaves near those frequencies. This has been corrected in this version.

From my point of view, the work can be published as is, but I confess that I don't see the results with the authors' initial enthusiasm. Please don't get me wrong.

The study is interesting but I'm not convinced with the solution compared to the anti-reflective films used for many years in solar panels and cameras. You have an
extra layer of material compared to the most common (and simple) designs I know. Therefore, the solution presented in the paper is more complex. For me it would be natural
to make a comparison with those other main stream systems, given their relevance in "current" technology and given the similarity with the authors' work.

This is the reason why I classified your paper as average when it comes to Scientific Soundness, Interest to the readers and Overall Merit.

Apart from that aspect, that the authors present only their study and there is no effort to compare their work with the work of others, I can say that the
expressions in the paper seem correct from the calculations I made in my previous revision. I especially enjoyed reading the introduction but I also enjoyed reading your paper.
All the best for you and good luck.

Author Response

 We made a minor changes  adding  a paragraph about a comparison with a current anti-reflection technology

Reply II to the referee 1.

Following the referee suggestions, we have specifically compared our results to those obtained from other mainstream systems and given their relevance in the context of current technology.

The following text was inserted at the end of the paper.

In conventional anti-reflective coatings, for example those used industrially in solar panels and cameras, normally an insulating SiO2 plate is covered by a graded SiO2/TiO2 film that acts as a destructive quarter wavelength interference layer. The other frequent methods for producing anti-reflection coatings utilise complex surface structures similar to moth-eye nano-tubules[1] or a combination of multilayers [2,3]. Such surfaces are often determined empirically to achieve the most optimal anti-reflection performance. The inclusion of anti-reflective coatings improves the efficiency of existing optoelectronic devices by minimizing light-loss due to reflection and removing glint. As a rule, in telescopes lenses multilayer coatings are preferred[2,3] to eliminate stray light from stellar observables. The moth-eye nanostructured anti-reflection coatings are often used in light-emitting diodes[4], displays[5], photovoltaic solar cells[6] and micro solar sensors[7]. The described technologies where the reflectivity is reduced through optical destructive interference at best give transmission rates of about 90%[8].  

Conventional anti-reflection approaches[6 -8] are not entirely suitable to the lower frequency THz and the microwave regimes. The issues arise because such coatings must be greater than the half-wavelength of incident light[9,10], which limits the size and range of the existing microwave and  THz devices [11]. In particular, the same problem occurs in anti-reflection technology for microwave and  THz transmission lines where, so far, various impedance matching transformershave been employed[12,13].

In the present paper, in our anti-reflective coating we have added an extra nano-meter thick metal layer of material compared to the most common (and simple) designs for anti-reflective films used for many years in diverse applications, e.g. in solar panels and cameras. We have discovered that due to this extra layer the light reflectivity drops to zero and we were able to achieve perfect impedance matching for several studied surfaces. We propose, that the method can be adopted to any existing designs, where the quality of anti-reflection can be increased to nearly 100% by depositing a very thin nm size metallic layer on top of the fabricated structures. The additional nm-metal coating does not practically increase the size (and cost) of the devices or affect its optical paths, but enables perfect impedance matching to occur widely across the electromagnetic spectrum, including in the microwave and THz ranges. These results may signal a new era in optoelectronic devices and open up novel avenues for diverse applications.

  1. Clapham, P. B. & Hutley, M. C. Reduction of lens reflexion by the “moth eye” principle. Nature 1973, 244, 281–282.

  1. Southwell, W. H. Gradient-index antireflection coatings. Opt. Lett. 1983, 8, 584–586.

  1. Grann, E. B., Varga, M. G. & Pommet, D. A. Optimal design for antireflective tapered two-dimensional subwavelength grating structures. J. Opt. Soc. Am. 1995, A 12, 333.

  1. Zhmakin, A. I. Enhancement of light extraction from light emitting diodes. Phys. Rep. 2011, 498, 189–241.

  1. Singh, R., Narayanan Unni, K. N. & Solanki, A. Improving the contrast ratio of OLED displays: an analysis of various techniques. Opt. Mater. 2012, 34, 716–723.

  1. Parida, B., Iniyan, S. & Goic, R. A review of solar photovoltaic technologies. Renew. Sust. Energ. Rev. 2011, 15, 1625–1636.

  1. Lee, C., Bae, S. Y., Mobasser, S. & Manohara, H. A novel silicon nanotips antireflection surface for the micro Sun sensor. Nano Lett. 2005, 5, 2438–2442.

  1. Yeh, P. Optical Waves in Layered Media. (Wiley, 2005).

  1. Wilson, S. J. & Hutley, M. C. The optical properties of “moth eye” antireflection surfaces. Opt. Acta. 1982, 29, 993–1009.

  1. Southwell, W. H. Pyramid-array surface-relief structures producing antireflection index matching on optical surfaces. J. Opt. Soc. Am. 1991, A 8, 549.

  1. Wang, L. J., Kuzmich, A. & Dogariu, A. Gain-assisted superluminal light propagation. Nature 2000, 406, 277–279.

  1. Klopfenstein, R. A transmission line taper of improved design. Proc. IRE 1956, 44, 31–35.

  1. Pozar, D. M. Microwave Engineering. (Wiley, 1997).

Reviewer 2 Report

Author made a good job in presentation and discussion of the results. I think it's better now and could be published. Only one remark is to reduce the caption of the figures which is too large and I think that part of the description could be included directly as a legend in the figure like the name of the layer, description of the coloured line, ...

Author Response

 We revised all  figures and made minor changes in figure captions to improve clarity